# Quasi-continuous melting of model polymer monolayers prompts reinterpretation of polymer melting

Ruibin Zhang[1,2,3], William S. Fall [1,4], Kyle Wm. Hall[5], Gillian A. Gehring[4], Xiangbing Zeng [2✉] & Goran Ungar [1,3✉]

Condensed matter textbooks teach us that melting cannot be continuous and indeed experience, including with polymers and other long-chain compounds, tells us that it is a strongly first-order transition. However, here we report nearly continuous melting of monolayers of ultralong n-alkane $C_{390}H_{782}$ on graphite, observed by AFM and reproduced by mean-field theory and MD simulation. On heating, the crystal-melt interface moves steadily and reversibly from chain ends inward. Remarkably, the final melting point is 80 K above that of the bulk, and equilibrium crystallinity decreases continuously from ~100% to <50% prior to final melting. We show that the similarity in melting behavior of polymers and non-polymers is coincidental. In the bulk, the intermediate melting stages of long-chain crystals are forbidden by steric overcrowding at the crystal-liquid interface. However, there is no crowding in a monolayer as chain segments can escape to the third dimension.

[1] State Key Laboratory for Mechanical Behavior of Materials, Shaanxi International Research Centre for Soft Materials, Xi'an Jiaotong University, Xi'an, China. [2] Department of Materials Science and Engineering, University of Sheffield, Sheffield, UK. [3] Department of Physics, Zhejiang Sci-Tech University, Hangzhou, China. [4] Department of Physics and Astronomy, University of Sheffield, Sheffield, UK. [5] Department of Chemistry, Temple University, Philadelphia, PA, USA. ✉email: x.zeng@sheffield.ac.uk; g.ungar@sheffield.ac.uk

Crystal melting is normally a strongly first-order transition. At the melting point ($T_m$), a pure substance faces a stark choice between being a low-energy, low-entropy crystal or a high-energy, high-entropy liquid (Fig. 1a). In some cases 3D long-range positional and orientational order of a crystal may disappear in two or more discrete intermediate steps; these are the liquid-crystalline states exhibited by some compounds, usually amphiphiles of anisometric shape (Fig. 1b)[1]. Most crystalline long-chain molecules and polymers do not form liquid crystals, but it has been suggested that their crystal-melt transition may not be sharp either: lamellar crystals may melt from the surface inward nearly continuously (Fig. 1c)[2-5]. However, only a very limited amount of "premelting" has actually been seen[6], involving disordering of a surface layer only a few atoms deep. Polymers usually melt over a range of temperatures but mainly due to the diversity in thickness of their lamellar crystals. When the thickness is uniform and thickening during heating is suppressed, melting occurs within 2 °C[7], and in long-chain monodisperse n-alkanes within a 1 °C interval[8]. However, here we report that nearly continuous equilibrium melting from the surface inward, as in Fig. 1c, can indeed occur in monolayers of long-chain compounds.

Our investigation centers on the melting behavior of the longest exactly monodisperse alkane ever synthesized, n-$C_{390}H_{782}$[9]. Monodisperse linear paraffins with chain length $n > 100$ have been synthesized through a series of protection-coupling-deprotection steps[8-11], and have provided unique model polymers allowing stringent tests of theories of polymer crystallization and morphology, without the obfuscations that arise from polydispersity[8,12-14]. Herein the melting behavior of graphite-supported monomolecular layers of n-$C_{390}H_{782}$ is investigated using atomic force microscopy (AFM). Large-scale molecular dynamics (MD) simulations are also performed, and a semiquantitative analytical model is constructed. Together these studies provide experimental evidence as well as an explanation of the extraordinary nearly continuous melting, ending 80 K above the bulk melting point of this model polymer. Our findings help understand the underexplored fundamentals of the phenomenon of polymer melting.

n-Alkanes $C_nH_{2n+2}$ and polyethylene (PE) molecules adhere particularly strongly to the graphite (001) surface due to a close epitaxial match between hydrogens on alternative $CH_2$ groups of an all-trans alkane chain (0.254 nm) and the centers of the six-membered rings of graphite (0.246 nm) (Fig. 2a, b). However, it should be noted that highly ordered lamellar structures have also been found in alkane monolayers adsorbed on atomically flat $MoSe_2$ and $MoS_2$ surfaces, even without such a special epitaxial relationship as that between alkanes and HOPG[15]. Previous studies on adsorbed short alkanes, mainly in the length range $20 < n < 40$[16-23] have found that monolayers on graphite melt at

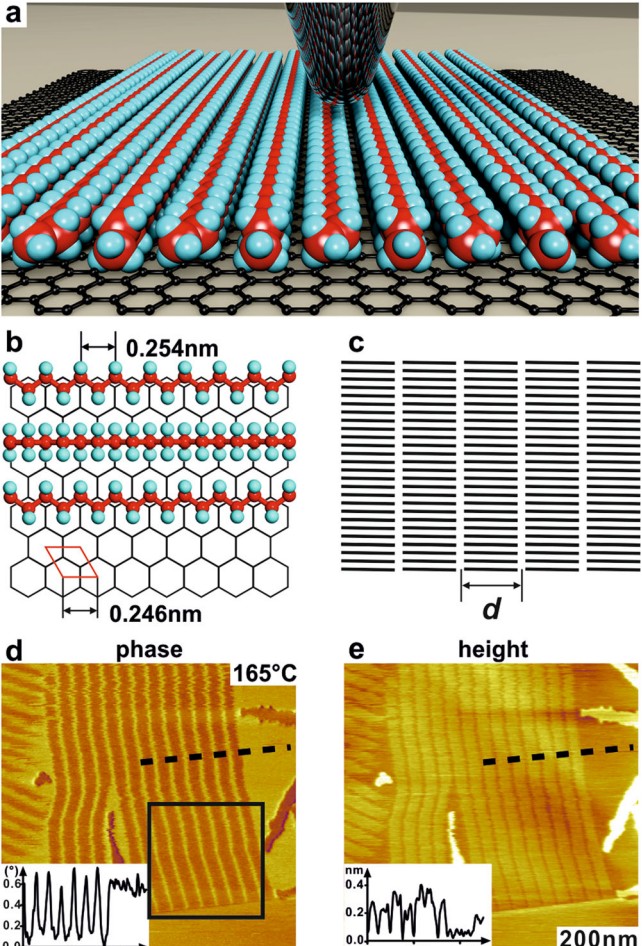

**Fig. 2 Schematic models and AFM phase images of a n-$C_{390}H_{782}$ monolayer on graphite. a** An artist's impression of an alkane monolayer on a graphite surface being probed by AFM. The sharpness of the AFM tip is not realistically presented for visual appeal. **b** Epitaxial arrangement of n-alkane chains on the graphite (001) surface. **c** 5 lamellae of 20 chains (schematic). **d, e** AFM phase and height images of a monolayer of n-$C_{390}H_{782}$ at 165 °C, with the respective phase and height profiles along the dotted lines shown in the insets. The black square in (**d**) marks the region shown magnified at different temperatures in Fig. 3.

temperatures a few degrees above the bulk melting point $T_m^{bulk}$, with additional phase transitions in the ordered state[24]. A "pre-freezing" effect several degrees above $T_m^{bulk}$ was also found in polyethylene thin layers[25]. Most notably, melting of monolayers 45 K above $T_m^{bulk}$ has been reported for n-$C_{60}H_{122}$[26]. However, there have been only three reported AFM studies of ultra-long alkanes[19,27,28].

## Results

**AFM characterization of n-$C_{390}H_{782}$ monolayers on graphite.** Figure 2d, e shows AFM phase and height images of a melt-crystallized monolayer film of n-$C_{390}H_{782}$ on graphite (not on a top surface of thicker adsorbate), with a thickness of a single alkane molecule (~0.4 nm, inset, Fig. 2e), recorded at 165 °C, more than 30 K above its $T_m^{bulk}$. Ribbons of n-alkanes can be clearly seen in both phase and height images. In the phase image (Fig. 2d), the bright lines (or edges of the ribbons) are soft amorphous regions around chain ends with increased phase lag $\phi$ in response to AFM tip tapping. The dark bands are stiffer ordered regions containing parallel chains which are known to

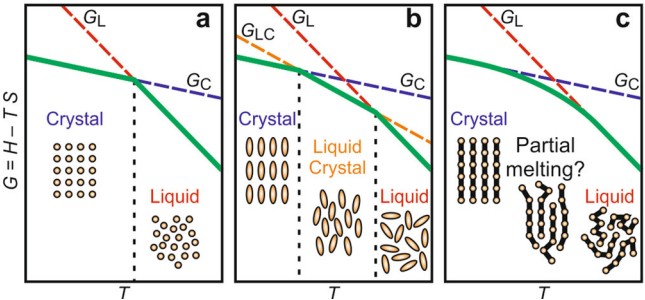

**Fig. 1 Schematic free energy-temperature profiles for crystal-liquid phase transition. a** A normal crystal. **b** A liquid-crystalline compound. **c** Hypothetic scenario for long-chain molecules/polymers. C, LC, and L are crystal, liquid crystal, and liquid phases.



**Fig. 3 Continuous melting of a $C_{390}H_{782}$ monolayer on graphite. a–e** Zoomed-in AFM phase images in the region marked by the black square in Fig. 2a, at temperatures from 120 to 211 °C. $d = 50.0$ nm.

align epitaxially along one of the three [100] directions of the graphite lattice[24,27,28]. According to previous studies, the orientation of the C–C–C zigzag plane of the all-*trans* alkane alternates from chain to chain between parallel and perpendicular to the graphite surface (Fig. 2b)[21].

By analogy with the structures seen in bulk alkanes and polymers we will refer to the monolayer ribbons of parallel chains as lamellae, to the regions around chain ends as lamellar surfaces, and to periodicity $d$ as lamellar thickness (Fig. 2c). The lamellar thickness is measured as 50.0 nm and, as expected, is very close to the 49.8 nm calculated chain length of n-$C_{390}H_{782}$.

Two extraordinary phenomena are observed on heating. Firstly, the lamellae are found to persist to above 211 °C (Fig. 3e), 80 °C above $T_m^{bulk}$ of n-$C_{390}H_{782}$, which is 132 °C[8]. Secondly, the soft stripes gradually widen with increasing temperature at the expense of the darker crystalline layers (Fig. 3a–e). Close to the final melting point $T_m$ their thickness even exceeds that of the remaining crystalline layers, with periodicity, $d$, remaining constant. The same behavior is also seen on cooling, hence the observed surface melting is an equilibrium phenomenon. We also calculated the averaged phase profiles $-\Delta\phi$ at different temperatures—see Fig. 4e; the originally narrow soft layers (minima) broaden to become eventually wider than the rigid middle before all periodicity disappears at $T_m$. Complementary grazing-incidence small-angle diffraction experiments on $C_{390}H_{782}$ monolayer on graphite have so far been unsuccessful due to extremely low contrast.

It has been previously reported that the melting point of thin PE films on Si, Al, and polyimide substrates are depressed rather than elevated[29], but the thickness of the films reported there are still above 10 nm or more, so the lamellar structure, though confined, is still 3D. In a more recent paper, however, it has been observed that there is a thin crystalline layer that is already stable above the bulk melting temperature at the melt/substrate interface for PE on Si and graphite surfaces[25], in line with our observations.

**MD simulations**. MD simulations of a $C_{390}H_{782}$ monolayer on graphite were performed as described in Methods and in Section 2.1, Supplementary Information. The experimental $\Delta\phi$ is proportional to the irreversible movement of chains by the AFM tip, which is impeded if the chains are straight and parallel as in the crystalline portion, but becomes allowed as the chain ends become misaligned. Hence we calculate the local alignment order parameter $P_2 = \frac{1}{2}\langle 3\cos^2\alpha - 1\rangle$ for all chain segments. Higher (lower) $P_2$ indicates higher (lower) segment order (for details see Section 2.1, Supplementary Information).

Maps of the simulated lamellar structure are provided in Fig. 4a based on the $P_2$ order parameter where the darker color corresponds to a larger value of $P_2$. The light areas reveal differing degrees of melting near the chain ends below $T_m$. Above $T_m$ all periodicity is lost. This is an interesting case of an almost 2D liquid as the alkane chains are effectively pinned down to the graphite surface by interaction between some of the alkane

hydrogens and the electron-rich graphite. However, some chain crossing and extension into the 3D can occur, an example being shown in Fig. 4d.

The $P_2$ maps indeed show a reasonable likeness to the AFM images. At $T > 102$ °C the order-disorder interfaces are evidently rough, with molecular centers displaced at random. It is known that even in the bulk, chain sliding is facilitated by moving $g^+tg^-$ kinks[30], so it must occur readily in a monolayer where out-of-plane displacement by kinks is allowed. At 77 °C the low-$P_2$ regions are rather localized (Fig. 4b), they are dispersed more widely with increasing temperature until bent chains predominate in the melt (Fig. 4c at 327 °C).

The match between the simulated $P_2$ order parameter maps and AFM phase images is best shown in Fig. 4e, f, where averaged $P_2$ profiles of the $C_{390}H_{782}$ lamellae at different temperatures are plotted together with the experimental phase profiles. In both cases the thickness of the ordered (crystalline) regions decreases with increasing temperature. At 211 °C almost half of the lamellar thickness (47%) is disordered, matching the experimental results. There are small but noticeable differences in the shapes of the AFM phase and MD order parameter profiles, particularly close to the melting temperature. These are attributed to their differences in spatial (~2 nm for AFM images and ~1 nm for MD simulations) and time resolution (MD simulations were sampled over a 5 ns window), and possible nonlinear responses from AFM tip in imaging.

The ordering of the molecular segments in MD simulations is also examined by calculating the average z-displacement away from the graphite substrate, as well as the chain mobility parameter (section 2.1, Supplementary Information). Both observables show trends similar to that of $P_2$ as a function of temperature. It is remarkable that aspects of molecular simulation, such as $T_m$, order parameters and average profiles are in such close agreement with experiment. One aspect where the simulation results differ from experiment is in the appearance of chain tilt (Fig. 4a). We note, however, that although in the high temperature range covered by this report experiments show no tilt, in fact at lower temperatures chains are indeed tilted, as will be elaborated in a forthcoming paper that will also report results on alkanes with a series of chain lengths[31].

**Mean-field model**. A mean-field model of a lamella of extended chains is constructed in order to explain the continuous thinning of the lamellae leading to their final melting. The model of an alkane chain with $n$ $CH_2$ (methylene) groups is shown in Fig. 5a where $x$ methylenes are molten; $x$ can be distributed freely between the two ends of the chain. However, a clean-cut switch between crystalline and molten states of the methylenes at the crystal-melt boundary is unrealistic. Conformations of chain segments emerging from the crystalline layer will be highly restricted due to the lack of lateral space, referred to as "overcrowding" at the crystal-melt interface (Fig. 5a)[32,33]. For mathematical expediency, in our model this is taken into account by

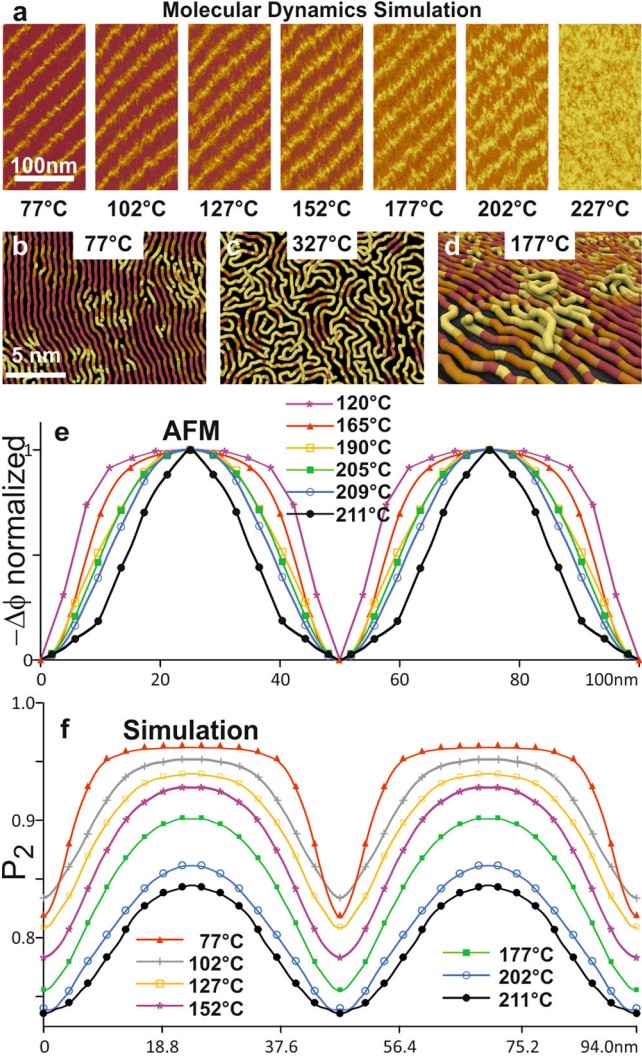

**Molecular Dynamics Simulation**

77°C  102°C  127°C  152°C  177°C  202°C  227°C

100nm

**b** 77°C  **c** 327°C  **d** 177°C

5 nm

**e AFM**

— 120°C
— 165°C
— 190°C
— 205°C
— 209°C
— 211°C

$-\Delta\phi$ normalized

0    20    40    60    80    100nm

**f Simulation**

$P_2$

1.0

0.9

0.8

— 77°C       — 177°C
— 102°C      — 202°C
— 127°C      — 211°C
— 152°C

0    18.8    37.6    56.4    75.2    94.0nm

**Fig. 4 Crystallinity maps and AFM phase shift profiles. a** Crystallinity ($P_2$) maps from MD simulations for temperatures between 77 and 227 °C (for details see section 2.1, Supplementary Information). Each map was calculated by partitioning the simulated surface into a 1 nm × 1 nm grid, and then locally averaging $P_2$ values across 51 instantaneous configurations (at 100 ps intervals). **b**, **c** Magnified snapshots of the simulations at 77 and 327 °C produced using OVITO[47]. The tubes correspond to individual alkane chains, and are colored according to their $P_2$ values. **d** A 3D MD simulation snapshot, taken at 177 °C, showing how chain segments detach from the substrate and cross over each other to resolve the overcrowding problem at the crystal-melt boundary in thin film. **e** Averaged and normalized AFM phase shift ($-\Delta\phi$) profiles of $C_{390}H_{782}$ at temperatures from 120 to 211 °C, with maximum phase shift at the surface (as 0) and minimum (as 1) at the center of the lamellae. **f** $P_2$ profiles calculated from the respective simulated $P_2$ maps. In our MD simulations the abscissa values are measured along the molecular direction which was actually inclined to the observed stripes.

assuming that the first $t/2$ groups of the melted ends, coming out of the crystalline layer, have the energy of the melt but the entropy of the ordered chain. Therefore, we can write the free energy of the partially melted chain as

$$F_P(x) = \begin{cases} xT_m^0 S_m - k_B T \ln(x+1) & 0 < x < t \\ x(T_m^0 - T)S_m + tTS_m - k_B T \ln(x+1) + 2\sigma_e & t < x < n-1 \end{cases}$$
(1)

Here $T_m^0$ is the ultimate melting point of the polymer of infinite length, $S_m$ is the melt entropy per $CH_2$, and $\sigma_e = H_e - TS_e$ is the

free-energy contribution from the two ends of each chain. The chain sliding entropy equals $k_B \ln(x+1)$, as for the $x$ end methylenes there are $x+1$ choices in selecting the position at which to separate the two ends by the $n-x$ ordered units. It is possible that one end of the chain can be longer than $t/2$ when $x < t$, but this is ignored in our treatment for mathematical simplicity. This simplification should not significantly affect much of our conclusions drawn below, as it will be shown that $t$ is as small as 1. When the surface is rough, i.e., when $x > t$, there is an additional increase in the system's free energy $2\sigma_e$ due to the disappearance of the smooth end surface (the factor 2 is used as each molecule has two ends).

The minimum free energy of a partially melted chain is therefore found at

$$x = x_0 = \begin{cases} \dfrac{k_B}{S_m}\dfrac{T}{T_m^0} - 1 & 0 < x_0 < t \\ \dfrac{k_B}{S_m}\dfrac{T}{T_m^0 - T} - 1 & t < x_0 < n-1 \end{cases}$$
(2)

This results in a high free-energy transition layer of thickness $t/2$ between crystal and melt. Thus the more severe the overcrowding the larger the $t$. However, in our model, partial melting of the chain does not in itself reduce the system free energy below $T_m$, as the entropy increase on melting the end segments comes with an increase in enthalpy due to loss of crystallinity. Stabilization of partially melted chains should come from the extra translational entropy $k_B \ln(x+1)$ of the molecules free to slide through the crystalline layer.

The equations allow us to evaluate the free energies of perfectly ordered chains, partially melted chains with $x < t$, with $x > t$, and of full melt. By equating pairs of free energies, we were able to reproduce the continuous partial melting and the final melting, all shown in Fig. 5b for n-$C_{390}H_{782}$, for different values of $t$ (for more details see section 2.2, Supplementary Information). At low temperatures crystallinity is nearly 100%, decreasing continuously with increasing temperature, and reaching around 50% before final melting, with the best-fit value of the "overcrowding" parameter $t = 1$ (red curve). For $t = 0$ (no overcrowding) melting becomes completely continuous. However, an increase in $t$ decreases $T_m$, and continuous melting disappears when $t \geq 8$.

## Discussion

In the following we compare melting in 2D and 3D systems of chain molecules and examine why no significant premelting occurs in the bulk. For perfectly aligned chains exiting a 3D crystal to instantly adopt random segment orientation, their effective cross-section must increase by a factor $g$ of 2–3 according to theory[32,33] and 2.3 according to experiment[34]. Therefore, in order for a polymer lamella to grow large laterally, a fraction $(1 - g^{-1})$ of all chains must either end at the interface, or reenter the crystal through a sharp fold to make room for the escaping chains. Regarding melting, there would be no thermodynamic advantage in a bulk lamella melting gradually by crystal thinning and thereby having to increase the number of energy-costly chain folds.

Interestingly, while not continuous, a two-step melting does actually occur in the bulk, in co-crystallizing binary mixtures of ultra-long alkanes[34]. A semicrystalline form (SCF, Fig. 5c) is stable within ~20 °C below the final $T_m$. Its lamellae contain fully crystalline shorter, and only partially crystalline longer chains, whose dangling ends form a liquid layer. As in the current part-melted monolayers, sliding entropy of the longer chains in the mixed binary alkane, due to the freedom of distribution of the $n-x$ dangling monomer units was also thought to stabilize the semicrystalline SCF phase. Another example of a nearly stable semicrystalline phase is the "non-integer form" (NIF) of pure long

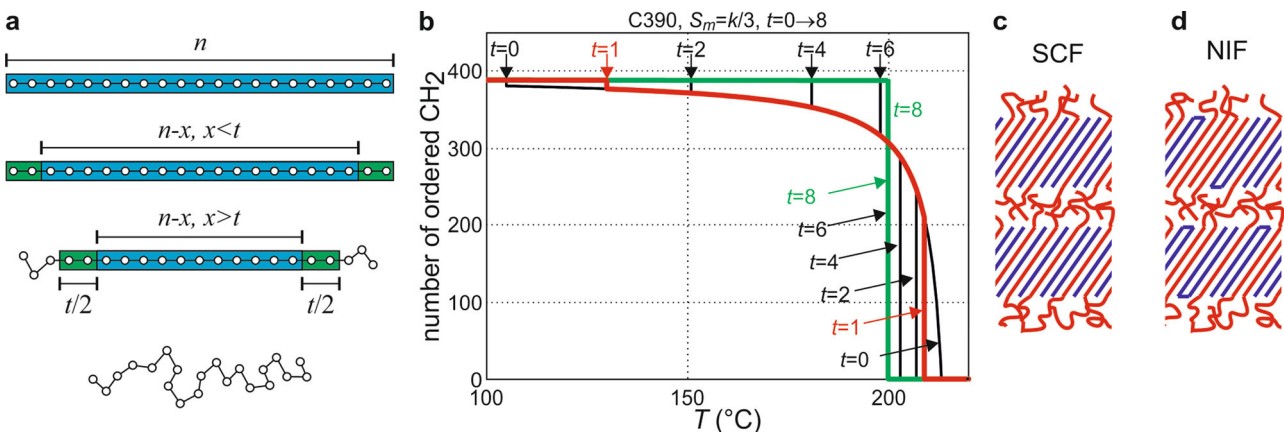

**Fig. 5 Mean-field theory and prediction of continuous melting. a** Schematic models of chains in different states. Top to bottom: fully crystalline, no surface disorder; smooth surface with small premelting/surface disorder, $x < t$; rough surface with significant premelting, $x > t$; and melt. **b** Number of $CH_2$ groups remaining in the ordered middle of the lamellae (proportional to crystallinity), calculated for transition layer thickness $t$ from 0 to 8. The temperature of the transition from smooth to rough surface (the small downward step in crystallinity) increases with increasing $t$, while the melting temperature of the lamellae (the final jump to 0 crystallinity) decreases. Consequently the temperature range of the rough surface and of partial melting becomes narrower, until for $t = 8$ it disappears completely. **c** Semicrystalline form (SCF) can be thermodynamically stable in binary mixtures of ultra-long n-alkanes with only the chain ends of the longer alkane melted (shorter alkane molecules are colored blue and longer alkane red)[34]. **d** Tight chain folding and tilting are needed to alleviate overcrowding and allow growth of metastable semicrystalline NIF lamellae in pure ultra-long alkanes (folded chain colored blue and non-folded red)[36].

alkanes, which crystallizes from melt below $T_m$ of the once-folded-chain form (Fig. 5d)[35,36]. NIF lamellae contain a mixture of highly crystalline chains folded precisely in half, and half-crystallized chains traversing the lamella only once and having dangling ends. NIF can only grow when enough chains are folded, freeing lateral space for the escaping half-crystallized chains. Both SCF and NIF are special cases of bulk systems where overcrowding is alleviated at one particular crystal thickness. Generally, surface overcrowding has been accepted as causing chain-folded crystallization of polymers, but its role in preventing continuous melting has not been recognized until now.

A key question to answer is why near-continuous melting is possible in a thin film but not in the bulk. Our results point to the following two reasons: Firstly and most importantly, the surface overcrowding problem is solved in thin film by "escape in the third dimension", with chain segments able to detach from the substrate and cross at the crystal-liquid interface, as evidenced by the simulation (Fig. 4d). This means a greatly reduced $t$ in 2D. An additional advantage of 3D escape is to facilitate $g^+tg^-$ kink formation and hence chain sliding, which stabilizes the semicrystalline state. The second factor deciding if melting could be continuous is the value of melt entropy $S_m$. For alkanes on graphite the best-fit has $S_m = k_B/3$. This low $S_m$ value is expected as molten chains are pinned to the substrate, making the melt nearly two-dimensional. When $S_m$ is lower, the number of disordered chain segments $x$ for the same undercooling $\Delta T$ is higher according to Eq. (2), as the entropy of chain sliding (proportional to $k_B \ln(x + 1)$) becomes relatively more effective in stabilizing the partially melted state. The higher the $S_m$, the less effective the sliding and the narrower the premelting range. However, if the overcrowding effect is small as in 2D ($t = 1$), taking the experimental value $S_m = 1.2k_B$ ($k_B/S_m = 0.83$) for bulk PE, the premelting range is narrower but still significant (Supplementary Fig. 1). This points to the presence of steric overcrowding at the crystal-amorphous interface being the dominant factor preventing continuous melting in the bulk.

In conclusion, as predicted long ago for polymers but never witnessed[2–8], continuous melting has now been observed for long-chain n-alkanes absorbed on graphite. Our experimental observations, MD simulation, and theoretical analysis show that

such behavior is a result of the much reduced overcrowding effect at the crystal-melt interface, and the much reduced entropy of an essentially 2D melt. The reversal of such conditions, particularly the presence of severe surface overcrowding, is the reason why premelting is almost negligible in bulk polymers and other chain systems. While in non-polymeric crystals all but the surface unit cells are thermodynamically equivalent and melt at the same temperature, in chain compounds the strong dependence of unit cells' free energy on the distance from lamellar surface would favor sequential melting, were it not for surface overcrowding that prevents it in the bulk.

The long alkane system studied here is different from polymers which contain many chain folds in their lamellae. While our study shows that chain sliding could contribute significantly to partial melting, it would be interesting to investigate, in the future, whether such partial melting behavior could be observed in polymers or model polymers containing a significant degree of chain folding and entanglement. It should be noted that such experiments will be complicated by the fact that chain folds are a nonequilibrium phenomenon, so it will be a challenge to separate kinetic and thermodynamic phenomena in the melting of lamellae with folded polymer chains. It would be interesting to explore the possible partial melting behavior of alkanes and other long-chain systems on other substrates as well.

## Methods

**Materials**. Long-chain n-alkane $C_{390}H_{782}$ was kindly provided by Dr. G. M. Brooke of Durham University. Highly ordered pyrolitic graphite (HOPG) wafers were obtained from Mikromasch, Germany. Ultrathin $C_{390}H_{782}$ films on HOPG were prepared by spin-coating, in which one droplet of alkane in toluene (1 mg/ml) was deposited on a freshly cleaved surface of HOPG at 2000 rpm with an Ossila spin coater. All sample films were subsequently dried in a vacuum oven for 2 h. Only data obtained on monolayers is included in the current report, even though multi-layers were also present in the samples prepared. The monolayer nature of the films reported here has been confirmed by checking the height difference between the film and the HOPG substrate, where the substrate was not fully covered by the sample. One example is shown in Fig. 2e.

**AFM**. The AFM experiment was performed in tapping mode using a Cypher ES AFM instrument, which was equipped with a heating stage allowing heating of sample in air from ambient to 250 °C in a sealed environment, with errors of less than 2 °C (more details see section 2.3 of Supplementary Information). AFM

scannings were carried out typically at 10 °C intervals on heating to detect changes in morphology. Precise phase transition temperatures were determined by imaging at 1 °C intervals. Scanning was carried out in repulsive force regime by keeping the phase shift below 90°, and a 90% set point ratio was used. The drive amplitude was gently increased for scanning at higher temperatures in order to prevent the tip from sticking to the sample surface.

The contrast in the tapping mode AFM height images of soft surface is determined by the real surface topography and tip-indentation. Deeper tip penetration on the soft amorphous regions contributes to the darker lines (lower apparent height) in the height image in Fig. 2e.

**MD simulations**. MD simulations were performed of a periodic 6-lamella monolayer structure composed of 1800 $C_{390}H_{782}$ molecules adsorbed on a graphite surface (141 nm × 281 nm). The system was modeled using the coarse-grain (CG) SDK model[37–39] without any alterations to the model. This CG model is appropriate for aliphatic and aromatic species[37–39], and has been previously verified to capture the important features of long-chain alkane systems and associated processes (e.g., crystallization)[40–42]. The CG SDK model represents 3 methylene units along an alkane backbone using a single CG segment (bead). All simulations were performed using the Large-scale Atomic/Molecular Massively Parallel Simulator[43] using common methodological strategies, and the prepared system was annealed from 77 to 327 °C in 25 °C intervals. The SDK model approximates graphene using a cubic representation. In keeping with previous studies on alkanes and polymers[40–42,44–46], the ordering of polymer chain segments (i.e., their crystallinity) was probed using the $P_2$ order parameter. More details can be found in section 2.1, Supplementary Information.

### Data availability

The data that support the findings of this study are available from the corresponding authors upon reasonable request.

### Code availability

Computer codes used in MD simulations of this study are available from the corresponding authors upon reasonable request.

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

## Acknowledgements

We are grateful to Professor Michael L. Klein at Temple University for providing the resources and support for the Molecular Dynamics Simulation work, and helpful discussions in paper preparation. We are most obliged to Drs. Gerald Brooke and Shahid Mohammed of University of Durham for providing the long alkane compound. This work used the ARCHER UK National Supercomputing Service (http://www.archer.ac.uk). The authors acknowledge funding from EPSRC (EP-P002250, EP-T003294), the 111 Project 2.0 of China (BP2018008), and from NSFC (grant 21674099).

## Author contributions

G.U. conceived and directed all aspects of the project. R.B.Z., supervised by G.U., carried out the AFM experiments. W.S.F., supervised by G.A.G. and K.W.H., carried out the MD simulations together with K.W.H. X.B.Z. developed the mean-field theory with support from G.A.G. All authors contributed to data analysis and production of tables and figures. X.B.Z. and G.U. prepared the manuscript with written contributions from all co-authors.

## Competing interests

The authors declare no competing interests.
