## [Peer Review File · Nature Communications]

REVIEWER COMMENTS

Reviewer #1 (Remarks to the Author):

The authors report their new finding of nearly continuous melting in the monolayer of long-chain n-alkane C390 through detailed investigations by AFM and MD simulation together with a mean field analysis. The present observation of the continuous and large-scale melting is quite in contrast to the sharp first order melting of usual materials and give valuable insight into the studies of melting, especially those in polymers. The expositions of the evidences, both by experiment and simulation, are very clear and I recommend publication of the manuscript in Nature Communication. Followings are a few points that would help to make the manuscript more convincing and easy to understand.

1. Though most of the researchers in this field of polymer science will understand it quite anomalous, proper citations of literatures related to the assertion (page 13) "as predicted long ago for polymers but never witnessed" will help to recognize the novelty of the present work.
2. The authors' claim (Page 6) "a remarkable likeness to the AFM image" is not always clear. In what sense are the MD image and the AFM image very alike? The correspondence of the transition temperatures is indeed remarkable. However, the lamellae in the AFM image (Fig. 3) seem considerably smooth compared to those of MD image (Fig.4a). In fact, the curves of the AFM and the MD data (Fig. 4 e and f) show appreciable difference in their slope at higher temperatures. I understand that there must be distinct differences in the spatiotemporal resolution, but some explanations about the apparent difference will be helpful.
3. The mean field analysis is very simple and clear but the interpretation of Fig.5b is not easily. Which path does the system follow, for example, in the case of large t value?
4. In the mean field model the authors consider the entropic contribution $\ln(x+1)$, which presumably comes from the positional freedom of the crystalline part of length $(n-x)$ in the chain. If I am right, I am afraid that one end of the chain can be longer than $t/2$ even if the summed length x and t satisfy $x < t$, then the equation (1) seems to have some problems.
5. Judging from the general knowledge of chain tilt (for example Figs. 5c), the present model for the MD simulation is also constructed to satisfy the chain tilt of about 45 (deg). However, in the present manuscript no explanation is given as to this point. Since the presence of large tilt has important contribution in relaxing surface overcrowding, relevant explanations would be favorable.
6. I think some corrections will be needed in the descriptions of Supplement 2.1.2.2. It is not readily understood what (the author write simply "It") is related to the AFM tip interacting with the sample. Also their "ASD" does not have proper physical dimension of squared length.

Reviewer #2 (Remarks to the Author):

This is a well written manuscript addressing a much discussed topic in polymer physics, namely the effect of confinement on polymer melting transitions. The authors simulate a long chain polymer via the synthesis of a super long chain alkane, with the argument that in this manner they overcome polydispersity issues common to polymer synthesis. The melting transition is then described via AFM measurements of the structure and morphology and confirmed via modeling.

There are several issues which I think should be addressed by the authors which would improve this manuscript;

First--the manuscript has intrinsic value as a study of melting in long alkanes where degrees of freedom associated with chain ends begin to matter. The association with polymers I believe detracts from their arguments. Their alkanes, long for alkane, are still short for practical polymer such as polyethylene, PE, which they refer to. Polymers had entanglements which add further constraints into their free energy functionals which are not considered here. Furthermore, polymer have short and long ranged lamellar ordering which the authors have not shown to be present in their alkanes.

As a result, the melting point in PE thin films was shown (Wang et al PRL 2006) to be depressed rather than elevated. I am not sure how to reconcile the results here with the previous publications, though I am sure there is an explanation based on the molecular structure, which is not immediately obvious. I believe the authors need to address this point, if they are drawing analogies to polymers.

Second--the authors show conformality between graphene and their alkane chains, but do not discuss surface interactions with their chains. Is graphite a strongly interacting surface or a completely non interacting surface. Polymer dynamics at surfaces are affected by pinning and unpinning which have to be considered. The influence of surface interactions on melting has been discussed by numerous groups where the melting point is known to be a function of the energy differential between the ordered and disordered state. The order imposed on the disordered state is a function of surface interactions and pinning and hence must be addressed in presentation of any theoretical model.

Third--the arguments presented regarding melting are based exclusively on AFM scanning. This technique is very sensitive to tip surface interactions, as well as to the dynamics of the study. In order to reach equilibrium, the samples must be annealed at their respective temperatures for adequate times to allow chain motion over the distances of the scan. This is very different than scattering techniques which sense immediate changes in fluctuation amplitudes and length scales that are more indicative of the immediate effects of melting. Hence have the authors considered glancing incidence scattering studies of their samples to corroborate their conclusions?

Reviewer #3 (Remarks to the Author):

The paper "Reinterpreting Polymer Melting: Insights from Quasi-Continuous Melting of Model Polymer Monolayers" by R. Zhang et al presents he interesting research of ultra-long alkanes behavior in ultrathin films on graphite substrate. Monitoring of lamellar organization of C390H782 alkane on graphite was performed at high temperatures and transformation and disappearance of the lamellar order was considered as the manifestation of lamellar melting. These observations were

analyzed with molecular dynamics simulations and the several conclusions of specific melting behavior of alkanes were made. This work is a good example of extraction of novel knowledge about melting phenomena of alkane layers in the interplay between experiment and theory and definitely will be motivating for researchers dealing with behavior of organic materials in confined geometry. In general, I am in favor of publishing this manuscript with the corrections that will address several comments presented below.

High-resolution visualization of normal alkane lamellar organization on this substrate, which was demonstrated in STM images in the pioneer publication in 1990 (G. C. McGonigal et al Appl. Phys. Lett. 1990, 57, 28), has been reproduced in large number of publications. Furthermore, self-assembly of alkane molecules in two dimensional lamellar order was also detected on MoSe₂ and MoS₂ substrates (e.g. S. Cincotti and J. P. Rabe "Self-assembled alkane monolayers on MoSe₂ and MoS₂" Appl. Phys. Lett. 1993, 62, 3531), which differ from graphite in lattices and nature of top surface atoms. This finding hints that a match of the lattice constants of the substrate and periodicities of zigzag structures of all-trans conformation of alkane chains does not matter for the formation of crystalline-like order of alkanes on atomically-flat substrates. This consideration should be taken into account by the authors who strongly advocate alkane epitaxy to graphite based on the match of adsorbate/substrate periodicities (Figure 2b). The artist's impression of the AFM probing of alkane/graphite samples (Figure 2a) is confusing as the shown tip size is abnormally smaller than H-atom. In practice the tip apex is 5-10 nm in diameter and this size is small enough for detection of the lamellar edges in ordered alkane layers.

As regarding the AFM experiments it will be instructing to expand their description by (1) explaining that the AFM images were recorded on single alkane layer on graphite and not on a top surface of thicker adsorbate; by (2) providing details of AFM heating experiments that insure correct temperature measurements that conducted with identical temperatures of the probe and the sample; by (3) presenting tip-sample amplitudes defining the tip-sample force interactions and explanation of height and phase contrast at the lamellar edges: the lamellar edges, which are formed by -CH₃ end groups, should look raised compared to chain core consisting of -CH₂- groups.

We are grateful for the reviewers' careful reading of the manuscript and for their expertly and pertinent comments. We agree with all of them and have made corresponding changes in the manuscript, as detailed point by point below.

Please see below our point to point reply, in italic, to Referees' comments.

Reviewer #1 (Remarks to the Author):

The authors report their new finding of nearly continuous melting in the monolayer of long-chain n-alkane C390 through detailed investigations by AFM and MD simulation together with a mean field analysis. The present observation of the continuous and large-scale melting is quite in contrast to the sharp first order melting of usual materials and give valuable insight into the studies of melting, especially those in polymers. The expositions of the evidences, both by experiment and simulation, are very clear and I recommend publication of the manuscript in Nature Communication.

We would like to thank the referee for their positive comments on our manuscript.

Followings area a few points that would help to make the manuscript more convincing and easy to understand.

1. Though most of the researchers in this field of polymer science will understand it quite anomalous, proper citations of literatures related to the assertion (page 13) "as predicted long ago for polymers but never witnessed" will help to recognize the novelty of the present work.

We have added references (2-8) at the end of the sentence, according to the referee's suggestion.

2. The authors' claim (Page 6) "a remarkable likeness to the AFM image" is not always clear. In what sense are the MD image and the AFM image very alike? The correspondence of the transition temperatures is indeed remarkable. However, the lamellae in the AFM image (Fig. 3) seem considerably smooth compared to those of MD image (Fig.4a). In fact, the curves of the AFM and the MD date (Fig. 4 e and f) show appreciable difference in their slope at higher temperatures. I understand that there must be distinct differences in the spatiotemporal resolution, but some explanations about the apparent difference will be helpful.

We agree with the referee in his comment that "a remarkable likeness" is hard to quantify and we have therefore changed the statement to "a reasonable likeness". With respect to comparisons between Fig. 3 and Fig. 4a (i.e., the experimental and computational surface images), the AFM images have a lower lateral resolution (~2nm) in comparison to that of the MD simulation (~1nm). It is worth noting that the MD simulations correspond to smaller systems and offer a "zoomed in" view on the lamellae compared to the experimental images. This can be seen by comparing the 100 nm scale bars in the respective images. Furthermore, the maps from the MD simulations were sampled over a 5 ns window, which is much shorter than the timescale used to acquire the experimental AFM images. As such, the slightly rougher appearance of the MD surfaces is to be expected. Possible nonlinearity in the response of the AFM tip to the structural order parameter contributes to the differences in AFM and MD results as well.

We have added the following sentence in the manuscript (page 8): "There are small but noticeable differences in the shapes of the AFM phase and MD order parameter profiles, particularly close to the melting temperature. These are attributed to their differences in spatial (~2nm for AFM images and ~1nm for MD simulations) and time resolution (MD simulations were sampled over a 5 ns window), and possible nonlinear responses from AFM tip in imaging."

3. The mean field analysis is very simple and clear but the interpretation of Fig.5b is not easily. Which path does the system follow, for example, in the case of large t value?

We have changed Fig. 5b so the path for the largest t ($t=8$) is marked in green and is now clearer. The following sentences have been added to the figure legend: "The temperature of the transition from smooth to rough surface (the small downward step in crystallinity) increases with increasing t , while the melting temperature of the lamellae (the final jump to 0 crystallinity) decreases. Consequently the temperature range of the rough-surface and of partial melting becomes narrower, until for $t = 8$ it disappears completely."

4. In the mean field model the authors consider the entropic contribution $\ln(x+1)$, which presumably comes from the positional freedom of the crystalline part of length $(n-x)$ in the chain. If I am right, I am afraid that one end of the chain can be longer than $t/2$ even if the summed length x and t satisfy $x < t$, then the equation (1) seems to have some problems.

What the referee has pointed out is correct. However, this possibility is ignored in our derivation for mathematical simplicity. This simplification, however, should not affect much of our conclusions drawn from the theory, as t has turned out to be very small. The following sentences have been added to the manuscript: "It is possible that one end of the chain can be longer than $t/2$ even when $x < t$, but this is ignored in our treatment for mathematical simplicity. This simplification should not affect much of our conclusions drawn below, as it will be shown that t is as small as 1."

5. Judging from the general knowledge of chain tilt (for example Figs. 5c), the present model for the MD simulation is also constructed to satisfy the chain tilt of about 45 (deg). However, in the present manuscript no explanation is given as to this point. Since the presence of large tilt has important contribution in relaxing surface overcrowding, relevant explanations would be favorable.

Yes, chain tilting with respect to lamellar normal is important in partially relieving surface overcrowding in the bulk. Our AFM results show that the molecular chains are not tilted, which is in agreement with our assessment that in such monolayers overcrowding is not a problem. However, as will be reported in a forthcoming paper (in preparation), chain tilt of 30° is indeed observed at temperatures lower than covered in the current report. The chain tilt of 45° in our MD simulations is larger than that experimentally observed, but we did not expect to reproduce chain tilt accurately, as we are representing the graphite surface using a coarse-grain (CG) cubic lattice, and each CG bead in the alkane chain represents 3 CH₂ groups. Future atomic scale simulations will be needed for such a task. We would like to add that a 45° chain tilt has been seen before in coarse-grained molecular dynamics simulations, for example in a similar study by Binder et al (Nano Letters 2017, 17, 4924-4928), where the tilt angle was also found to depend on the length of alkane molecules due to "subtle interplay of the positional excess entropy ... and the entropy of chain bending fluctuations."

This being said, our current simulations do still recover the majority of the relevant phenomena associated with the long-chain alkane monolayers; the goal of our simulations was to provide additional qualitative insights into the monolayer melting process, which we do.

We have made the following modifications to our manuscript, page 8, after "... close agreement with experiment." One aspect where the simulation results differ from experiment is in the appearance of chain tilt (Figure 4a). We note, however, that although in the high temperature range covered by this report experiments show no tilt, in fact at lower temperatures chains are indeed tilted, as will be elaborated in a forthcoming paper that will also report results on alkanes with a series of chain lengths.³¹"

We have also added the following sentences in Supplementary Information 2.1.1: “Note that the 45° chain tilt is a result of the coarse grain nature of our MD simulations, and has been found in similar coarse-grained MD simulations previously.⁵¹ In particular, the polymer chains are represented using coarse-grain beads where a single coarse-grain bead represents three methylene, and graphene has been coarse-grained based on a cubic lattice approximation. For a faithful reproduction of chain tilt at lamellar surfaces, an atomic scale simulation will be needed in the future.”

6. I think some corrections will be needed in the descriptions of Supplement 2.1.2.2. It is not readily understood what (the author write simply "It") is related to the AFM tip interacting with the sample. Also their "ASD" does not have proper physical dimension of squared length.

Many thanks to the referee for spotting the mistakes in the equations and their descriptions in SI section 2.1.2.2. Instead of Average Squared Displacement, we were calculating the “Positional Standard Deviation (PSD)”, and it should be the square of PSD that is proportional to the force constant. The revised Supplemental Sec. 2.1.2.2 is shown below:

2.1.2.2 Mobility/ Positional Standard Deviation (PSD)

By calculating both the average squared position and average position, it is possible to quantify the movement of a given bead about its mean position throughout the simulation. The mobility (or PSD) for an arbitrary bead i , is defined as

$$PSD = \sqrt{\langle x_i^2 \rangle - \langle x_i \rangle^2 + \langle y_i^2 \rangle - \langle y_i \rangle^2 + \langle z_i^2 \rangle - \langle z_i \rangle^2}$$

Where the angle brackets indicate averaging over 51 simulation snapshots in a given 25ps time window during the simulation, corresponding to 0.5ps intervals. It may be related to the force of the AFM tip interacting with the sample by equipartition assuming that the tip is a harmonic oscillator with Hamiltonian.

$$H = \sum_i \frac{p_i^2}{2m} + \frac{k_x x_i^2}{2}$$

Where p_i is the momentum, k_x is the force constant and x_i is the position. By equipartition this is equivalent to

$$H = \langle H_{pot} \rangle + \langle H_{kin} \rangle = \frac{1}{2} k_B T + \frac{1}{2} k_B T$$

Hence we can show that the PSD^2 is proportional to the force constant at a given temperature in the simulation.

$$\frac{1}{k} \propto \frac{PSD^2}{k_B T}$$

Reviewer #2 (Remarks to the Author):

This is a well written manuscript addressing a much discussed topic in polymer physics, namely the effect of confinement on polymer melting transitions. The authors simulate a long chain polymer via the synthesis of a super long chain alkane, with the argument that in this manner they overcome polydispersity issues common to polymer synthesis. The melting transition is then described via AFM measurements of the structure and morphology and confirmed via modeling.

Thanks to the referee for their positive comments.

There are several issues which I think should be addressed by the authors which would improve this manuscript;

First--the manuscript has intrinsic value as a study of melting in long alkanes where degrees of freedom associated with chain ends begin to matter. The association with polymers I believe detracts from their arguments. Their alkanes, long for alkane, are still short for practical polymer such as polyethylene, PE, which they refer to. Polymers had entanglements which add further constraints into their free energy functionals which are not considered here. Furthermore, polymer have short and long ranged lamellar ordering which the authors have not shown to be present in their alkanes.

We agree with the referee on the difference between long alkanes and polymers. Incidentally, we note that the length of $C_{390}H_{782}$ is already twice the entanglement length for polymethylenes. We have added the following sentence in the conclusion section: "The long alkane system studied here is different from polymers which contain many chain folds in their lamellae. While our study shows that chain sliding could contribute significantly to partial melting, it would be interesting to investigate, in the future, whether such partial melting behaviour could be observed in polymers or model polymers containing a significant degree of chain folding and entanglement. It should be noted that such experiments will be complicated by the fact that chain folds are non-equilibrium, so it will be a challenge to separate kinetic and thermodynamic phenomena in the melting of lamellae with folded polymer chains."

As a result, the melting point in PE thin films was shown (Wang et al PRL 2006) to be depressed rather than elevated. I am not sure how to reconcile the results here with the previous publications, though I am sure there is an explanation based on the molecular structure, which is not immediately obvious. I believe the authors need to address this point, if they are drawing analogies to polymers.

The Wang et al results are different to ours in two ways. The "thin" film in Wang et al paper is still over 10nm, and compared to our monolayer alkane film (only ~0.4nm thickness) the lamellar structure and overcrowding effect in the former is still 3D and more similar to the bulk. The substrate used is different too, as stated in our manuscript, there is strong epitaxial relationship between graphite and alkane molecules, but not so much for the Si, Al and PI substrates, as also evidenced by the observed "flat on" lamellae in Wang's papers for their "thin" lamellae. We would like also to draw the referee's attention to a more recent paper (Löhmann et al, PNAS, 2014, original ref. 23 in our manuscript), where it has been observed that there is a thin crystalline layer that is already stable above the bulk melting temperature at the melt/substrate interface for PE on Si and graphite surfaces.

We have added reference to the Wang 2006 paper: "It has been previously reported that the melting point of thin PE films on Si, Al and Polyimide substrates are depressed rather than elevated,²⁸ but the thickness of the films reported there are still above 10nm or more, so the lamellar structure, though confined, is still 3D. In a more recent paper, however, it has been observed that there is a thin crystalline layer that is already stable above the bulk melting temperature at the melt/substrate interface for PE on Si and graphite surfaces, in line with our observations.²⁴"

Second--the authors show conformality between graphene and their alkane chains, but do not discuss surface interactions with their chains. Is graphite a strongly interacting surface or a completely non interacting surface. Polymer dynamics at surfaces are affected by pinning and unpinning which have to be considered. The influence of surface interactions on melting has been discussed by numerous groups where the melting point is known to be a function of the energy differential between the ordered and disordered state. The order imposed on the disordered state

is a function of surface interactions and pinning and hence must be addressed in presentation of any theoretical model.

The graphite surface is highly commensurate with an alkane chain, and there is strong interaction between one out of the four C_2H_4 hydrogens with the electron-rich core of the carbon hexagons in the graphite (see also references 15-26 in our manuscript). Hence alkane molecules are strongly "pinned" down on the graphite surface. The sentence starting "This is an interesting case of an almost 2D liquid ..." is now changed to read: "This is an interesting case of an almost 2D liquid as the alkane chains are effectively pinned down to the graphite surface by interaction between some of the alkane hydrogens and the electron-rich graphite. However, some chain crossing and excursion into 3D can occur, an example being shown in Fig. 4d."

Third--the arguments presented regarding melting are based exclusively on AFM scanning. This technique is very sensitive to tip surface interactions, as well as to the dynamics of the study. In order to reach equilibrium, the samples must be annealed at their respective temperatures for adequate times to allow chain motion over the distances of the scan. This is very different than scattering techniques which sense immediate changes in fluctuation amplitudes and length scales that are more indicative of the immediate effects of melting. Hence have the authors considered glancing incidence scattering studies of their samples to corroborate their conclusions?

GISAXS and GIWAXS have indeed been attempted on monolayer on graphite but without success. The films are extremely thin (only $\sim 4 \text{ \AA}$), and lamellar diffraction is expected on the equator below the horizon and requires a nearly zero incidence angle beam hence giving a very weak signal. Furthermore, as expansion on melting takes place by escape into 3D rather than by in-plane expansion, the surface electron density actually remains unaffected and uniform, thus leaving the GISAXS contrast unchanged and very low. Neutron diffraction, using deuterated samples, is a theoretical possibility, but finding and financing qualified chemists to undertake the difficult, risky and costly year-long multi-stage synthesis of deuterated $C_{390}D_{782}$ is not a realistic proposition at present. We have added the following sentence to the end of the AFM section on page 5, "Complementary grazing-incidence small-angle diffraction experiments on $C_{390}H_{782}$ monolayer on graphite have so far been unsuccessful due to extremely low contrast."

Reviewer #3 (Remarks to the Author):

The paper "Reinterpreting Polymer Melting: Insights from Quasi-Continuous Melting of Model Polymer Monolayers" by R. Zhang et al presents the interesting research of ultra-long alkanes behavior in ultrathin films on graphite substrate. Monitoring of lamellar organization of $C_{390}H_{782}$ alkane on graphite was performed at high temperatures and transformation and disappearance of the lamellar order was considered as the manifestation of lamellar melting. These observations were analyzed with molecular dynamics simulations and the several conclusions of specific melting behavior of alkanes were made. This work is a good example of extraction of novel knowledge about melting phenomena of alkane layers in the interplay between experiment and theory and definitely will be motivating for researchers dealing with behavior of organic materials in confined geometry. In general, I am in favor of publishing this manuscript with the corrections that will address several comments presented below.

We thank the referee for their positive comments.

High-resolution visualization of normal alkane lamellar organization on this substrate, which was demonstrated in STM images in the pioneer publication in 1990 (G. C. McGonigal et al Appl. Phys. Lett. 1990, 57, 28), has been reproduced in large number of publications. Furthermore, self-

assembly of alkane molecules in two dimensional lamellar order was also detected on MoSe₂ and MoS₂ substrates (e.g. S. Cincotti and J. P. Rabe “Self-assembled alkane monolayers on MoSe₂ and MoS₂” Appl. Phys. Lett. 1993, 62, 3531), which differ from graphite in lattices and nature of top surface atoms. This finding hints that a match of the lattice constants of the substrate and periodicities of zigzag structures of all-trans conformation of alkane chains does not matter for the formation of crystalline-like order of alkanes on atomically-flat substrates. This consideration should be taken into account by the authors who strongly advocate alkane epitaxy to graphite based on the match of adsorbate/substrate periodicities (Figure 2b).

We agree with the referee that it would be interesting to examine other substrates and to see if the same or similar behaviour can be observed. We have added the following sentence in the manuscript: “It would be interesting to explore the possible partial melting behaviour of alkanes and other long chain systems on other substrates as well.”³⁶ We would like to thank the referee for pointing us to the APL 1990 paper, which has been added to our manuscript as the new reference 15.

The artist’s impression of the AFM probing of alkane/graphite samples (Figure 2a) is confusing as the shown tip size is abnormally smaller than H-atom. In practice the tip apex is 5-10 nm in diameter and this size is small enough for detection of the lamellar edges in ordered alkane layers.

We have modified the picture so the sharpness of the tip is less exaggerated, and added an extra explanation in the figure legend: “The sharpness of the AFM tip is not realistically presented for visual appeal.”

As regarding the AFM experiments it will be instructing to expand their description by (1) explaining that the AFM images were recorded on single alkane layer on graphite and not on a top surface of thicker adsorbate;

Page 4, last paragraph, the first sentence has been modified: “Figs. 2d,e show AFM phase and height images of a melt-crystallized monolayer film of n-C₃₉₀H₇₈₂ on graphite (not on a top surface of thicker adsorbate)”.

by (2) providing details of AFM heating experiments that insure correct temperature measurements that conducted with identical temperatures of the probe and the sample;

Description of the heating control in AFM experiment has been added as follows.

“The passive heat transfer and good insulation of the heating stage in Cypher ES Environmental AFM provide precise control of sample temperature (± 2 °C as calibrated using the melting of n-C₄₀H₈₂ and benzoic acid). During the scanning, the resonance frequency of the cantilever remained constant, indicating that thermal equilibrium has been reached between the AFM probe and the sample.”

by (3) presenting tip-sample amplitudes defining the tip-sample force interactions and explanation of height and phase contrast at the lamellar edges: the lamellar edges, which are formed by –CH₃ end groups, should look raised compared to chain core consisting of –CH₂- groups.

Our AFM experiments working in the tapping mode do not provide the resolution or sensitivity to catch the difference between CH₃ and CH₂ groups. To our knowledge, the direct observation of methyl groups with AFM has only been done where the –CH₃ groups were stabilised by the surrounding crystal structure e.g. in polypropylene crystals^{1,2} where steric hindrance stops the methyl side group being knocked around by the tip. In our case, however, the height contrast

between the crystalline interior and amorphous chain ends is determined by tip-indentation. The tip penetrates more in the soft amorphous regions and consequently its apparent height is lower. We have added extra explanation in the experimental section to explain the height contrast observed:

“The contrast in the tapping mode AFM height images of soft surface is determined by the real surface topography and tip-indentation. Deeper tip penetration on the soft amorphous regions contributes to the darker lines (lower apparent height) in the height image in Fig. 2e.”

We believe that the changes we made in response to the helpful and pertinent comments by all three reviewers have further improved the manuscript which, we hope, is now acceptable for publication.

1 . Uchida, K., Mita, K., Matsuoka, O., Isaki, T., Kimura, K., Onishi, H. *The structure of Uniaxially Stretched Isotactic Polypropylene Sheets: Imaging With Frequency-modulation Atomic Force Microscopy.* *Polymer*. **82**, 349-355 (2016).

2. Kocun, M., Labuda, A., Meibold, W., Revenko, I., Proksch, R. *The structure of uniaxially stretched isotactic polypropylene sheets: Imaging with frequency-modulation atomic force microscopy.* *ACS NANO*, **11**, 10097-10105 (2017).

REVIEWER COMMENTS

Reviewer #1 (Remarks to the Author):

I think the manuscript is now well-revised and can be recommended for publication as is. The author's finding that the reduced overcrowding at the lamella surface gives rise to the continuous melting with thinning is I think very insightful in many related problems besides those of 2D crystals.

Reviewer #2 (Remarks to the Author):

The authors have done a very good job of replying to all reviewers comments. I recommend publication. It is now a very well written and well rounded manuscript.

Reviewer #3 (Remarks to the Author):

In my review of the paper I mentioned that epitaxy of alkanes on HOPG as the reason of their lamellar order is not justified. Furthermore, similar order is observed for alkanes on other layered materials where there is not direct match between crystal lattices of the substrate and adsorbate. This point was not addressed in the revised version as in the authors' response they only limited by saying that the order on other substrates should be explored.

Second point of my earlier comments was regarding a temperature calibration in high-temperature AFM studies. In the authors' response they mentioned about the performed calibration using n-C₄₀H₈₂ alkanes (melting temperature 81C) and benzoic acid (melting temperature 122C). These temperatures are substantially lower than those used in studies of C₃₉₀H₇₈₂, and I doubt that the made calibration justifies the correct values of much higher applied temperatures. The mentioned stability of the probe resonance as the proof of thermal equilibrium does not mean that temperatures of the probe and sample were identical.

Finally, I don't see the evidence that single alkane layers used in the experiments. The wide range of concentrations between 1 and 10 mg/ml of C₃₉₀H₇₈₂ solution in toluene, which was documented in Methods, suggests thickness variability of alkane adsorbates on HOPG.

To summarize, I am still in favor of publishing the manuscript and will appreciate if the authors brings more details and discussion regarding the above-mentioned issues.

We thank the reviewers for their comments and suggestions for revision. Our reply to Reviewers are in *Italic* below.

Reviewer #1 (Remarks to the Author):

I think the manuscript is now well-revised and can be recommended for publication as is. The author's finding that the reduced overcrowding at the lamella surface gives rise to the continuous melting with thinning is I think very insightful in many related problems besides those of 2D crystals.

Reviewer #2 (Remarks to the Author):

The authors have done a very good job of replying to all reviewers comments. I recommend publication. It is now a very well written and well rounded manuscript.

Many thanks to Reviewers 1 and 2 for their positive comments on our manuscript.

Reviewer #3 (Remarks to the Author):

In my review of the paper I mentioned that epitaxy of alkanes on HOPG as the reason of their lamellar order is not justified. Furthermore, similar order is observed for alkanes on other layered materials where there is not direct match between crystal lattices of the substrate and adsorbate. This point was not addressed in the revised version as in the authors' response they only limited by saying that the order on other substrates should be explored.

We did not claim in our manuscript that "epitaxy of alkanes on HOPG" is the reason for the existence of the highly ordered alkane monolayer at a substrate surface, as the reviewer seems to suggest. We have moved the reference to the observation by STM of ordered adsorbed alkane monolayers on MoSe₂ and MoS₂ substrates (original ref. 37) to the last paragraph of our introduction, and added further explanation to the work to make this point clearer.

The last paragraph of the introduction of our manuscript has been changed to:

"n-Alkanes C_nH_{2n+2} and polyethylene (PE) molecules adhere particularly strongly to the graphite (001) surface due to a close epitaxial match between hydrogens on alternative CH₂ groups of an all-trans alkane chain (0.254nm) and the centers of the six-membered rings of graphite (0.246nm) (Figs. 2a,b). However, it should be noted that highly ordered lamellar structures have also been found in alkane monolayers adsorbed on atomically flat MoSe₂ and MoS₂ surfaces,¹⁵ without such epitaxial relationship as between alkanes and HOPG. ..."

Second point of my earlier comments was regarding a temperature calibration in high-temperature AFM studies. In the authors' response they mentioned about the performed calibration using n-C₄₀H₈₂ alkanes (melting temperature 81C) and benzoic acid (melting temperature 122C). These temperatures are substantially lower than those used in studies of C₃₉₀H₇₈₂, and I doubt that the made calibration justifies the correct values of much higher applied temperatures. The mentioned stability of the probe resonance as the proof of thermal equilibrium does not mean that temperatures of the probe and sample were identical.

Ideally we should have calibrated the heating stage at higher temperatures, closer to the ultimate melting point of our long alkane, for example by checking the melting of tin. Unfortunately the instrument, including the heating stage control, is not in proper working order any more with no

prospect of being fixed any time soon, so we cannot carry out this calibration now. However, please see below the reply we received from Asylum Research about the heating stage calibration:

“The PolyHeater stages are calibrated across their full range of temperature at the factory and this calibration is then stored as coefficients on a flash memory chip on the PolyHeater itself. We typically see errors of <2 degrees with a temperature probe reading the top surface of the stage that has been well-coupled to the heater stage.”

Their statement of errors of <2 degrees is in line with our own calibration at lower temperatures using C₄₀H₈₂ and benzoic acid. We have also checked the bulk melting of C₃₉₀H₇₈₂ on HOPG, using the same setup, and the measured melting point was 130.5°C, compared to its 132.0°C melting temperature in literature.

We have modified the second paragraph of the AFM methods section as follows:

“The passive heat transfer and good insulation of the heating stage in Cypher ES Environmental AFM provide precise control of the sample temperature. The stage is calibrated across the full range of temperatures at factory, with stated errors of less than 2 °C at the top surface of the stage. This is in line with our own calibration using the melting point of n-C₄₀H₈₂ (measured 81.0°C, 82.0°C in literature), benzoic acid (measured 121.0°C, 122.3°C in literature), and bulk C₃₉₀H₇₈₂ (measured 130.5°C, 132.0°C in literature). HOPG substrate is a very good thermal conductor, and the mass of an alkane monolayer on top of the HOPG surface is extremely small. Therefore we expect only a small deviation in the sample temperature from that measured by the instrument. During the scanning, the resonance frequency of the cantilever remained constant, indicating that thermal equilibrium had been reached between the AFM probe and the sample. While a small temperature difference between the sample and the AFM probe cannot be completely excluded, the apex of the AFM probe has rather small thermal mass and there will be some thermal resistance between the AFM probe and sample given the tip geometry. Moreover the AFM has been operating in the tapping imaging mode throughout, thus the probe was always in intermittent contact with the sample surface.”

Finally, I don't see the evidence that single alkane layers used in the experiments. The wide range of concentrations between 1 and 10 mg/ml of C₃₉₀H₇₈₂ solution in toluene, which was documented in Methods, suggests thickness variability of alkane adsorbates on HOPG.

The stated concentrations between 1 and 10 mg/ml were mistakenly included from a previous version of the manuscript where different alkanes and multi-layers were investigated and it escaped our attention in modification. For C₃₉₀H₇₈₂ monolayer films, the lowest 1 mg/ml concentration was used. This has now been corrected. Only data obtained on monolayers were included in the current manuscript, even though multi-layers were also present in the samples prepared. The monolayer nature of the films reported in our manuscript was confirmed by AFM height images at places where the alkane film did not fully cover the HOPG substrate, and one of such height images, together with the height profile, is shown in Figure 2e of the manuscript.

The materials section has been changed to:

“Ultrathin C₃₉₀H₇₈₂ films on HOPG were prepared by spin-coating, in which one droplet of alkane in toluene (1 mg/ml) was deposited on a freshly cleaved surface of HOPG at 2000 rpm with an Ossila spin coater. All sample films were subsequently dried in a vacuum oven for 2 hours. Only data obtained on monolayers were included in the current report, even though multi-layers were also present in the samples prepared. The monolayer nature of the films reported here was confirmed by

checking the height difference between the film and the HOPG substrate, where the substrate was not fully covered by the sample. One example is shown in Figure 2e.”

To summarize, I am still in favor of publishing the manuscript and will appreciate if the authors brings more details and discussion regarding the above-mentioned issues.

Many thanks to Reviewer 3 for their positive feedback and helpful suggestions.

REVIEWERS' COMMENTS

Reviewer #3 (Remarks to the Author):

The latest revision of the paper, in which all raised questions were properly addressed, can be published.